# Keeping an Eye on Perimenopausal and Postmenopausal Endometriosis

**DOI:** 10.3390/diseases7010029

**Published:** 2019-03-12

**Authors:** Michail Matalliotakis, Charoula Matalliotaki, Alexandra Trivli, Maria I. Zervou, Ioannis Kalogiannidis, Maria Tzardi, Ioannis Matalliotakis, Aydin Arici, George N. Goulielmos

**Affiliations:** 1Department of Obstetrics & Gynecology, Venizeleio General Hospital of Heraklion, 71409 Crete, Greece; charoulamat@outlook.com.gr (C.M.); matakgr@yahoo.com (I.M.); 23rd Department of Obstetrics and Gynecology, Aristotle University of Thessaloniki, 54124 Thessaloniki, Greece; ikalogiannidis@gmail.com; 3Section of Molecular Pathology and Human Genetics, Department of Internal Medicine, School of Medicine, University of Crete, 71003 Heraklion, Greece; alextrivli@yahoo.com (A.T.); zervou@uoc.gr (M.I.Z.); goulielmos@med.uoc.gr (G.N.G.); 4Histopathology Department, University General Hospital of Heraklion, Medical University of Crete, 71110 Heraklion, Greece; tzardi@med.uoc.gr; 5Department of Obstetrics & Gynecology and Reproductive Sciences, Yale University School of Medicine, New Haven, CT 06510, USA; aydin.arici@yale.edu

**Keywords:** endometriosis, perimenopause, postmenopause, dry eye

## Abstract

**Introduction**: We aimed to describe and review the epidemiological aspect of the disease pattern of a series of perimenopausal and postmenopausal women with a histology confirmation of endometriosis. **Material and Methods**: We retrospectively examined the clinical records of 184 perimenopausal and 46 postmenopausal women with endometriosis. Data were collected and analyzed from 1100 patients’ charts with confirmed endometriosis and involved cases from two different geographical areas, New Haven (US) and Greece. The statistical methods included ×^2^ and the Mann-Whitney U test. In the perimenopausal group (age 45–54 years), there were 184 patients (16.7%) and the postmenopausal group (55–80 years) had 46 (4.2%). The average age of diagnosis was (49 ± 2.3) and (61.2 ± 5.1), respectively (*p* < 0.01). **Results**: Advanced endometriosis was more aggressive in the perimenopausal group (*p* < 0.05); in the same group, we observed a higher left-sided predisposition of endometriosis in comparison with the right side (*p* < 0.01). Endometrioma was the most common gynecological condition among patients with perimenopausal endometriosis in relation to the postmenopausal group (*p* < 0.001). Additionally, we found uterine leiomyomata more prominent in the perimenopausal group (*p* < 0.05). In contrast, adenomyosis was found higher in postmenopausal patients (*p* < 0.05); further, 24 cases with dry eye we observed. **Conclusions**: Postmenopausal endometriosis is an important underestimated condition. Although the reported situation is not common, various clinicopathological characteristics were observed in both groups. Clinicians should be aware that there is a correlation between endometriosis and endometriosis-associated ovarian cancer in perimenopausal and postmenopausal age.

## 1. Introduction

Endometriosis is an estrogen-dependent chronic inflammatory condition, estimated to occur in 5%–10% of women of reproductive age [1]. Perimenopausal and postmenopausal endometriosis is rare, because of the reduction or absence of estrogen hormone production, which could prevent estrogen–dependent endometriosis [2].The human perimenopausal period, which starts when ovarian function begins to decline, is defined by the World Health Organization and the North American Menopause Society as a period of 2–8 years preceding menopause and lasts up to one year after the last menstruation [3,4]. Although endometriosis typically ends when menopause occurs due to the decreased levels of estrogens, reactivation of endometriosis can occur in some postmenopausal women, either as a side effect of therapy with administered hormones or because of the presence of endogenous hormones [5,6]. The prevalence in the postmenopausal group varies from 2% to 5% [7].

A PubMed search with the key word “endometriosis” yielded a total of 24,678 papers. By using the combination of “endometriosis’’ and “postmenopause”, we detected 347 papers or 0.7% of the total literature on endometriosis. Moreover, 64 case reports (18%) on postmenopausal endometriosis were observed (accessed January 2018). In postmenopause, the real incidence of endometriosis is limited and not consistent. Recently, Haas et al. performed a retrospective analysis of 42,079 women with surgically confirmed endometriosis in Germany and they found 1,074 (2.55%) cases in the climacteric age group (55–95 years) [8]. Moreover, in a subsequent study, Haas et al. [9] found that physical fitness and freedom from physical restrictions, a good social environment and psychological care in the postmenopausal period lead to a remarkable improvement in pain, dyspareunia, and influence on sexual life in women with endometriosis.

In the framework of this study, we aimed to investigate the clinical characteristics of a series of perimenopausal and postmenopausal women with endometriosis, to present any relation between other gynecologic conditions in our cohort and, furthermore, to describe the disease pattern in postmenopausal cases.

## 2. Materials and Methods

The current work involved 1,100 women with endometriosis from two different countries, over a 25-year period. The clinical, surgical and pathological records of 550 women with endometriosis, who underwent surgical treatment between 1996–2005 at Yale University Hospital and 550 cases between 1990 and 2015 from the Departments of Obstetrics and Gynecology at the University of Crete and Venizeleio General Hospital of Crete, were reviewed. The stage of endometriosis was scored according to the revised Classification of the American Fertility society [10].

The clinicopathologic features of the gynecologic diseases were classified according to the criteria of FIGO [11]. In our inclusion criteria, we involved all cases confirmed by tissue biopsy and laparotomy or laparoscopy during the perimenopausal and postmenopausal period. The primary exclusion criteria were patients with incomplete medical records and surgically induced endometriosis.

Data, which included age, symptoms, stage and side of endometriosis, were recorded. Information on the histological type of cancer and gynecologic conditions were obtained from pathological records. Group I included 184 patients in perimenopause, and group II had 46 women with postmenopausal endometriosis.

The Human Committee of Yale University School of Medicine approved this study (HIC #12590) as did the Ethics Committee for Human Research of Venizeleio Hospital (ECHR#46/6686 and 47/773/2017).

Student t test and ×^2^ tests were used for comparison of the mean of the various characteristics. The Mann-Whitney U test was performed for data not distributed normally. The results are reported as mean ± SD or as percentages where appropriate. Differences were considered statically significant at *p* < 0.05.

## 3. Results

During the study period, 1,100 medical records with a diagnosis of endometriosis or endometrioma were identified and reviewed. We found 184 (16.7%) patients with perimenopausal endometriosis (age 45–54 years) and 46 (4.2%) patients with postmenopausal endometriosis (55–80 years). Mean age at the time of surgery was different for perimenopausal women (49 ± 2.3) compared to the postmenopausal group with endometriosis (61.2 ± 5.1) (*p* < 0.01) (Table 1). In the postmenopausal cases, we reported 6 women in the age of 75 to 80 years old. The main symptoms were similar in the two groups (Table 1). However, advanced endometriosis (stage III, IV) was more aggressive in the perimenopausal group (*p* < 0.05). (Table 1). Furthermore, in patients with perimenopausal endometriosis, we found left-sided endometriosis in 98/152 (64.5%), compared to right-sided 54/152 (35.5%) (*p* < 0.01) (Table 1). In the postmenopausal group, the observed proportion of endometriosis was similar in both sides (left and right) (Table 1).

Table 2 shows the results of distribution of gynecologic conditions between the two groups. Endometrioma was the most common condition among women with perimenopausal endometriosis, 125/184(68%) in comparison with the postmenopausal group 5/46 (10.8%) (*p* < 0.001). Besides, we found uterine leiomyomata more frequent in the perimenopausal group (*p* < 0.05); in contrast, adenomyosis was found to be higher in postmenopausal patients (*p* < 0.05). Endometriosis-associated ovarian cancer and uterine cancer was similar in both groups (Table 2). In addition, according to the medical records, 24 women in the postmenopausal group complained of dry eye symptoms (*p* < 0.001).

## 4. Discussion

In the current work, we investigated the epidemiological aspect of perimenopausal and postmenopausal endometriosis. Our results support a significant association between endometriosis in the perimenopausal and postmenopausal age and further confirm the coexistence of endometriosis with numerous benign or malignant diseases.

A PubMed search of the literature in the English language found that a few studies were focused on postmenopausal endometriosis. In 1960, Kempers et al. conducted the first large study that described postmenopausal endometriosis. Over a 13-year period, they observed 39 cases with postmenopausal endometriosis [12]. Morotti et al. studied the clinical records of 72 postmenopausal cases and reported that the median age was 58.5 years. Moreover, they concluded that the usual location of endometriotic lesions was the ovary [2].

In general, endometrioma represents the most frequent adnexal masses in the premenopausal population [13]. During the perimenopausal period, estrogen is gradually decreased; thus, in our retrospective study, we detected more endometrioma cases in the perimenopausal group compared to the postmenopausal group.

Interestingly, according to Haas et al., clinicians should take into consideration the risk of endometriosis in cases of unclear pelvic pain in the perimenopausal age [8]. In a recent study, Anastasi et al. observed the coexistence of endometriosis with moderate pelvic pain and a significant decrease of 25-OH-vitamin D levels; this phenomenon might be due to the imbalance of prostaglandin production, resulting in an altered inflammatory cascade [14].

Depriest et al. suggested a correlation between postmenopausal endometriosis and ovarian cancer [15]. In our case, we found 184 patients (16.7%) in the perimenopause group and 46 (4.2%) in the postmenopausal group. Even though these data are consistent with the observations of previous studies and indicate that physicians should be aware of the possibility of endometriosis and endometriosis-associated ovarian cancer in postmenopausal age, we cannot tell if cancer and endometriosis coincide, thus posing a limitation to our study. A review of the literature confirms that this is the first large report to examine the association of endometriosis in perimenopausal and postmenopausal age.

Although the mechanism through which endometriosis may be associated with postmenopausal age is not clear, several theories are described. During the reproductive stage, it is well known that endometriosis represents an estrogen-dominated syndrome [16,17]. Estrogen production during menopause may be derived from extra-ovarian sources such as the adrenal gland, endometrial stroma, adipose tissue, and skin [18]. Other possibility is that of hormonal replacement therapy (HRT), which can reactivate endometriosis or even create new implants in climacteric women with history of the disease [16]. The effect of postmenopausal hormonal therapy on recurrence of postmenopausal endometriosis is contradictory.

In 2010, the European Menopause and Andropause society postulated that hormonal therapy may reactivate residual implants, and further increase the risk of malignant transformation in postmenopausal women [19,20,21]. In our findings, we noticed a greater incidence of uterine leiomyomata in the perimenopausal group, although adenomyosis was prominent in postmenopausal patients. Adenomyosis appears during the reproductive period; thus, our findings are not in agreement with the literature—it is believed that adenomyosis is a pre-existing condition that can diagnosed via surgery after menopause [22].

Of note, hormonal disturbance can affect the appropriate function of distinct organs. We detected several cases with dry eye in the postmenopausal group. This can be explained by two mechanisms. First, age is responsible for reduced tear production by the lacrimal gland [23]. Moreover, the endocrine system plays a significant role in the regulation of the ocular surface and the development and/or treatment of aqueous-deficient and evaporative dry eye disease (DED). Both estrogen and progesterone affect the anatomy and physiology of the lacrimal gland and tear secretion, can been detected in human tears, and are reported to be correlated with levels in serum of premenopausal females [24]. Finally, hormonal replacement therapy (HRT) alone has been associated with the prevalence of DED [25].

Considering that an important genetic component accounts for both endometriosis [26] and the variability observed in the timing of menopause [27], we assumed that it would be reasonable to review genetic factors that have been reported to influence both perimenopausal and postmenopausal endometriosis in order to extract evidence for clinical recommendations to handle this disease. Although these conditions are rare, it is important to be aware of a putative genetic background that may contribute to the development of these conditions, especially for postmenopausal endometriosis, which infers a risk of malignant transformation [6]. To date, it has been observed that aromatase inhibitors treat both premenopausal and postmenopausal endometriosis by suppressing local estrogen production. Notably, postmenopausal women produce estrogen by the steroidogenic gene aromatase, which is expressed in the stromal cells of endometriosis [28]. Furthermore, no evidence for an association between CYP1B1 genotype and postmenopausal endometrial cancer risk was found in a study performed by Rylander-Rudqvistet et al. [29]. Importantly, it has been reported that, with regard to perimenopausal endometriosis, endometrioid ovarian carcinomas (ENOC) and clear cell ovarian carcinomas (CCOC) are associated with mutations of *ARID1A,* a tumor suppressor gene [30], *PTEN*, *KRAS* and -catenin (*CTNNB1*) genes [31]. However, no genetic data regarding the predisposition or prediction for perimenopausal and postmenopausal endometriosis have been collected thus far.

## 5. Conclusions

We aimed to describe and compare the demographic characteristics and disease features of perimenopausal and postmenopausal cases with endometriosis. Our results support an association between endometriosis with perimenopausal and postmenopausal age. We detected that endometrioma, uterine leiomyomata, and advanced endometriosis are more prominent in the perimenopausal group, compared to the postmenopausal group, where adenomyosis was found at a higher rate. We also confirmed an association between endometriosis and endometriosis-associated ovarian cancer in both groups; thus, clinicians should keep an eye out for follow up on a long-term basis in cases operated for endometriosis.

## Figures and Tables

**Table 1 diseases-07-00029-t001:** Clinical characteristics and frequency of left- and right–sided unilateral ovarian endometriosis in the two groups.

	Group 1 Perimenopausal Women with Endometriosis (*n* = 184)	Group 2 Postmenopausal Women with Endometriosis (*n* = 46)	*p*-Value
**I. Age (years)** at time of surgery	49 ± 2.3 (45–54)	61.2 ± 5.1 (55–80)	*p* < 0.01
**II. Main complaints (%)**			
Pelvic pain	34(18.4%)	10(21.8%)	N.S
Adnexal mass	150(81.6%)	36(78.2%)	N.S
**III. Endometriosis stage (%)**			
Stage I	20(11%)	21(45.6%)	*p* < 0.05
Stage II	23(12.5%)	15(32.6%)	*p* < 0.05
Stage III	69(37.5%)	5(10.9%)	*p* < 0.05
Stage IV	72(39%)	5(10.9%)	*p* < 0.05
**IV. Side predisposition**			
Left-sided	98/152(64.5%)	20/38(52.6%)	N.S
Right-sided	54/152(35.5%)	18/38(47.4%)	N.S
Bilateral	32	8	
All unilateral	*n* = 152	*n* = 38	
	Left/right *p* < 0.01	Left/right N.S	

Data are expressed as the mean ± SD or Percentage as appropriate; N.S = Not Significant.

**Table 2 diseases-07-00029-t002:** Thecoexistence of endometriosis with various benign and malignant conditions (*n* and %).

	Group 1 Perimenopausal Women with Endometriosis (45–54 years) *n* = 184	Group 2 Postmenopausal Women with Endometriosis (55–80 years) *n* = 46	*p*-Value
1. Endometrioma	125(68%)	5(10.8%)	*p* < 0.001
2. Ovarian or Paraovarian cyst	21(11.4%)	7(15.2%)	N.S
3. Uterine Leiomyomata	82(44.5%)	8(17.4%)	*p* < 0.05
4. Adenomyosis	29(16%)	15(32.6%)	*p* < 0.05
5. Uterine leiomyomata and adenomyosis	21(11.4%)	5(10.8%)	N.S
6. Other benign gynecologic conditions	13 (7%)	3 (6.5%)	N.S
7. EAOC (Endometriosis associated ovarian cancer)	8 (4.3%)	3 (6.5%)	N.S
8. Uterine cancer	6 (3.2%)	6 (13.1%)	N.S
9. Bowel cancer	3 (1.65)	0	N.S
10. Other cancers	2 (1%)	1 (2.1%)	N.S
11. DED (Dry eye disease)	0	24 (52.1%)	*p* < 0.001
Total	184/1100(16.7%)	46/1100(4.2%)	

Data are expressed as the mean ± SD or Percentage as appropriate; NS = Not Significant.

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
