# Peer review of "Keeping an Eye on Perimenopausal and Postmenopausal Endometriosis"

_diseases, 2019, doi:10.3390/diseases7010029_

Round 1
Reviewer 1 Report
Overall the paper is suitable for the field of the journal and has an interesting topic to study further but in its present form presents several critical issues and requires major re-write.
There’s a need for significant structural changes to improve readability and make scientific content more coherent.
Abstract
- In conclusions part of Abstract and Conclusions section it is mentioned hormonal therapy but neither in materials and methods nor in Results sections is reported that the population in study is treated with hormonal therapy; additionally it is not specified the type (i.e. GnRHa, progestogens or combined estrogen-progestogens) or duration of therapy.
-The aim of the study is not well expressed, and incomplete in comparison to the aim described in the manuscript.
-It is reported that the risk of EAEOC is just in postmenopausal women whereas in “Results” section and in Table 2 it is stated that it has the same incidence also in perimenopausal women.
Introduction
-I suggest to investigate other factors influencing endometriosis associated pain such as 25 OH Vitamin D status, as reported in this recent article “ Anastasi E, Fuggetta E, De Vito C, Migliara G, Viggiani V, Manganaro L, Granato T, Benedetti Panici P, Angeloni A, Porpora MG. Low levels of 25-OH vitamin D in women with endometriosis and associated pelvic pain. Clin Chem Lab Med. 2017 Oct 26;55(12):e282-e284”
-The reference Gonzales (2009) should be better discussed.
Material and methods
-The size of the two is too heterogeneous
-Precise inclusion and exclusion criteria should be provided.
Results
-Non statistically significant results shouldn’t be discussed.
-It is not clear why bowel cancer and other types of cancer are included in the table described as “gynecologic conditions”.
Discussion
The Discussion section should start with key result of the study and after that authors should comment and discuss their results in comparison with results of similar studies. I suggest to the authors to rewrite this whole section because there are parts that are not directly associated with this study or should be included in Introduction not Discussion.
-It is not clear why among the different clinical manifestations of menopause you chose to discuss about DED (dry eye disease), since in your study you didn’t find any correlation with endometrosis and DED is mentioned only as one of the main indication for hormonal replacement therapy. (lines 144-151)
-Additionally, the genetic mutations reported link perimenopause and postmenopause to other clinical conditions (cardiovascular disease or spinal bone loss) than endometriosis, which is the object of the study. (lines 159-163).
-Lines 126; 163-164, the claims should be corroborated by references.
-It could be interesting to discuss the underlying mechanism for the presence of adenomyosis in postmenopausal endometriosis and to explain why endometrioma is mostly present in perimenopausal endometriosis.
-There is a lack of figures and the tables are poorly designed and don’t report all the discussed data (i.e. in table 2 dried eye symptoms are not reported, even if discussed)
Conclusion
-All conclusions should be based solely on the obtained result which, in this study, is not the case.
-The sentence “ postmenopausal endometriosis infers a risk of malignant transformation such as endometriosis associated ovarian cancer especially in postmenopausal patients.” doesn’t make sense.
-It has not been demonstrated that postmenopausal endometriosis infers a risk of malignant transformation because in the Results section it is stated that the incidence of EAOC is similar into groups.
-You should specify which type of association you found between endometriosis and perimenopausal and postmenopausal age.
-Moreover, the effects of hormonal replacement therapy in patients with previous endometriosis are mentioned again, but these results were not demonstrated in this study.

Author Response
Response to Reviewer 1 Comments
Comments and Suggestions for Authors
1. Overall the paper is suitable for the field of the journal and has an interesting topic to study further but in its present form presents several critical issues and requires major re-write.
There’s a need for significant structural changes to improve readability and make scientific content more coherent.
Response: Authors are grateful to the reviewer for the positive and encouraging comments. We have revised our present research paper in the light of his/her useful suggestions and comments.
2. Abstract
- In conclusions part of Abstract and Conclusions section it is mentioned hormonal therapy but neither in materials and methods nor in Results sections is reported that the population in study is treated with hormonal therapy; additionally it is not specified the type (i.e. GnRHa, progestogens or combined estrogen-progestogens) or duration of therapy.
Response: This statement is excluded.
-The aim of the study is not well expressed, and incomplete in comparison to the aim described in the manuscript.
Response: We aimed to describe and review the epidemiological aspect of the disease pattern of a series of perimenopausal and postmenopausal women with a histology confirmation of endometriosis.
-It is reported that the risk of EAEOC is just in postmenopausal women whereas in “Results” section and in Table 2 it is stated that it has the same incidence also in perimenopausal women.
Response: We further evaluated the suggestion, in the revised version we included both groups.
In the revised version, the conclusion section of the abstract has been deleted and rephrased.
3. Introduction
-I suggest to investigate other factors influencing endometriosis associated pain such as 25 OH Vitamin D status, as reported in this recent article “ Anastasi E, Fuggetta E, De Vito C, Migliara G, Viggiani V, Manganaro L, Granato T, Benedetti Panici P, Angeloni A, Porpora MG. Low levels of 25-OH vitamin D in women with endometriosis and associated pelvic pain. Clin Chem Lab Med. 2017 Oct 26;55(12):e282-e284”
Response: we included the suggestion of the reviewer in the discussion part.
4. -The reference Gonzales (2009) should be better discussed.
Response: The sentence has been changed and rephrased.
5. Material and methods
-The size of the two is too heterogeneous
Response: As suggested we modified this section.
-Precise inclusion and exclusion criteria should be provided.
Response: In our inclusion criteria we involved all cases confirmed by tissue biopsy and laparotomy or laparoscopy during the perimenopausal and postmenopausal period. The primary exclusion criteria were patients with incomplete medical records and surgically induced endometriosis.
Results
-Non statistically significant results shouldn’t be discussed.
Response: Non statistically significant results are limited in the tables.
-It is not clear why bowel cancer and other types of cancer are included in the table described as “gynecologic conditions”.
Response: As suggested, we tried to improve and modified the content of the table.
Table II. The coexistence of endometriosis with various benign and malignant conditions (n and %).
6. Discussion
The Discussion section should start with key result of the study and after that authors should comment and discuss their results in comparison with results of similar studies. I suggest to the authors to rewrite this whole section because there are parts that are not directly associated with this study or should be included in Introduction not Discussion.
Response: In the current work, we investigated the epidemiological aspect of the perimenopausal and postmenopausal endometriosis.
Our results support a significant association between endometriosis in the perimenopausal and postmenopausal age and further confirm the coexistence of endometriosis with numerous benign or malignant diseases.
Moreover, the discussion part has been deleted and rephrased as suggested.
-It is not clear why among the different clinical manifestations of menopause you chose to discuss about DED (dry eye disease), since in your study you didn’t find any correlation with endometrosis and DED is mentioned only as one of the main indication for hormonal replacement therapy. (lines 144-151)
Response: Well of note, a hormonal disturbance can affect the appropriate function of distinct organs. We detected several cases with dry eye in the postmenopausal group. This can be explained by two mechanisms. First, age is responsible for reduced tear production by the lacrimal gland [23]. Moreover; the endocrine system plays a significant role in the regulation of the ocular surface and the development and/or treatment of aqueous-deficient and evaporative dry eye disease (DED). Both estrogen and progesterone affect the anatomy and physiology of the lacrimal gland and tear secretion, can been detected in human tears, and are reported to be correlated with levels in serum of premenopausal females [24]. Finally, hormonal replacement therapy (HRT) alone has been associated with the prevalence of DED [25].
The increased number of cases (24/46(52.1%)) in the postmenopausal group impressed us, so we decided to include that in our epidemiologic aspect.
-Additionally, the genetic mutations reported link perimenopause and postmenopause to other clinical conditions (cardiovascular disease or spinal bone loss) than endometriosis, which is the object of the study. (lines 159-163).
Response: As suggested, this sentence was deleted.
-Lines 126; 163-164, the claims should be corroborated by references.
Response: We improved the suggestion.
-It could be interesting to discuss the underlying mechanism for the presence of adenomyosis in postmenopausal endometriosis and to explain why endometrioma is mostly present in perimenopausal endometriosis.
Response: As suggested, we explained the above recommendations in the discussion part.
-There is a lack of figures and the tables are poorly designed and don’t report all the discussed data (i.e. in table 2 dried eye symptoms are not reported, even if discussed)
Response: We tried to modify and further to improve the content of the tables, according to the guidelines of both reviewers .
7. Conclusion
-All conclusions should be based solely on the obtained result which, in this study, is not the case.
Response: We aimed to describe and compare the demographic characteristics and disease features of perimenopausal and postmenopausal cases with endometriosis.
Our results support an association between endometriosis with perimenopausal and postmenopausal age. We detected that endometrioma, uterine leiomyomata and advanced endometriosis are more prominent in perimenopausal group, compared to the postmenopausal group, where adenomyosis was found in a higher rate.
Apart from that, we confirmed an association between endometriosis and endometriosis- associated ovarian cancer in both groups; thus clinicians should keep an eye for a follow up on a long-term basis in cases operated for endometriosis.
-The sentence “ postmenopausal endometriosis infers a risk of malignant transformation such as endometriosis associated ovarian cancer especially in postmenopausal patients.” doesn’t make sense.
Response: As suggested, the sentence is rephrased.
-It has not been demonstrated that postmenopausal endometriosis infers a risk of malignant transformation because in the Results section it is stated that the incidence of EAOC is similar into groups.
Response: As suggested, we included both groups.
-You should specify which type of association you found between endometriosis and perimenopausal and postmenopausal age.
Response: We detected that endometrioma, uterine leiomyomata and advanced endometriosis are more prominent in perimenopausal group, compared to the postmenopausal group, where adenomyosis was found in a higher rate.
Apart from that, we confirmed an association between endometriosis and endometriosis- associated ovarian cancer in both groups; thus clinicians should keep an eye for a follow up on a long-term basis in cases operated for endometriosis.
-Moreover, the effects of hormonal replacement therapy in patients with previous endometriosis are mentioned again, but these results were not demonstrated in this study.
Response: this section was excluded according to the reviewers’ guidelines.

Reviewer 2 Report
This manuscript describes the patient demographics and disease features of 184 perimenopausal and 46 postmenopausal women with endometriosis, drawn from a population of 1100 women diagnosed with endometriosis in New Haven (USA) and Greece. The authors compared features between the peri- and post-menopausal women, and found some differences in disease presentation (e.g. perimenopausal women generally had higher stage endometriosis, and were more likely to have endometriomas and uterine leiomyomata, than postmenopausal women).
This paper is useful in providing data on endometriosis in a relatively under-studied population, as endometriosis is often viewed as a disease more of young, pre-menopausal women. The authors also discuss the important clinical concern that endometriosis carries a risk of malignant transformation, which other studies have suggested may be more common in older, peri- or post-menopausal women, and describe endometriosis-associated malignancies in their population.
General comments
Some general points which I think the authors may which to address are:
1. Rounding of numbers: Percentages in tables and in the text are mostly given to one decimal place, but on occasion are rounded to the nearest whole number. Ideally, one should be consistent (e.g. either round to the nearest whole number, or give to one decimal place, but do the same throughout).
2. Decimal points: The authors frequently use a comma (,) rather than a point (.) to separate decimal places (e.g. 18, 4% rather than 18.4%, or p<0,01 rather than p<0.01). While use of a comma in this situation is common in Europe, I believe it is less common in the USA, and so it might be clearer for an international audience to use a point.
3. Discussion: In addition to putting the findings of this study into the context of other papers (which the authors do here), it might also be helpful to discuss some potential limitations of the study. For example, to consider possible confounding or biases in the population of patients used here: these are cases identified from clinical chart review, and with a histological diagnosis of endometriosis. These women thus will not be entirely representative of all women with endometriosis in the community (e.g. they may have had some other reason to bring them to hospital and have surgery, such as the cancers or other gynaecological conditions described here). Hence, this study cannot, for example, provide data on the incidence of endometriosis-associated cancers amongst women with endometriosis in the general population.
4. Conclusion: It is not clear to me that the results described in this paper provide evidence for all the conclusions stated here. In some cases, I think this may simply be suboptimal phrasing. For example: “Our results support an association between endometriosis with perimenopausal and postmenopausal age”; I think the authors may simply mean: “Our results provide further evidence that endometriosis may occur in women of perimenopausal and postmenopausal age.” The authors then state: “Postmenopausal endometriosis infers a risk of malignant transformation such as endometriosis associated ovarian cancer especially in postmenopausal patients.” While it is true that endometriosis carries a risk of malignant transformation (as has been shown by previous papers), and some previous papers have suggested that the risk of malignant transformation may be higher in postmenopausal women (e.g. the paper by DePriest et al 1992), the potential confounding effect of age should perhaps be considered. Postmenopausal women are almost by definition older, and ovarian cancer is more common with increasing age. I’m not clear if there is any evidence that postmenopausal women with endometriosis have a higher rate of malignant transformation than younger women with endometriosis, after adjusting for age? If the authors are aware of such a study, it would be good to cite it (this reviewer may simply have missed it). Finally, the authors state “A larger study would be required to confirm the correlation of endometriosis and menopause”: I am not entirely certain what ‘correlation’ this refers to. If it is simple co-existence, then I think this has been shown (by this paper and others). The term ‘correlation’ perhaps more usually implies a statistical association, in which two factors increase or decrease together (e.g. a correlation between smoking and increasing rates of lung cancer, or between use of the oral contraceptive pill and decreasing rates of ovarian cancer), and I don’t think this paper shows or cites a correlation as such between endometriosis and the menopause. I think the authors may perhaps mean more generally ‘the relationship between endometriosis and menopause’?
5. References: the authors appear to have correctly numbered the references at the end of the manuscript in citation order, but have used ‘(Author, Date)’ to cite them within the text. However, I believe the reference style for this journal is to number the references in square brackets. Some references are also given in non-standard style, with variable underlining etc. This would need to be corrected prior to publication.
More minor specific comments
Various more minor specific comments are below. These are mostly to do with phrasing, and the authors may obviously modify as they think best.
1. Introduction, page 1, line 37: “reproductive female population” might be better phrased as “women of reproductive age”.
2. Introduction, page 2, line 45: “or because the presence of endogenous hormones” should perhaps be either “or because of the presence of endogenous hormones” or “or due to the presence of endogenous hormones”.
3. Introduction, page 2, line 51: “In postmenopause the real incidence of endometriosis is limited and not consistent” should perhaps be rephrased, e.g. “In postmenopausal women, evidence for the real incidence of endometriosis is limited and not consistent”.
4. Materials and Methods, page 2, lines 77-79: It is not entirely clear to me why some of this sentence is italicised. Perhaps this was not intentional?
5. Materials and Methods, page 2, lines 82-85: some of the phrasing of the ethics statement seems a little odd. Assuming this still accurately reflects the approvals given, might it be clearer to say: “This study was approved by the Human Committee of Yale University School of Medicine (HIIC #12590), the Ethics Committee for Human Research of Venizeleio Hospital (ECHR#46/6686 and 47/773/2017), the Department of Obstetrics and Gynecology of the University of Crete, and the Department of Obstetrics and Gynecology of the Venizeleio General Hospital of Crete.” [Though obviously please modify if necessary].
6. Results, page 3, line 95: I think a full stop (.) is missing at the end of one sentence and the beginning of the next (“(Table 1). In the…”).
7. Results, page 3, line 96: “we reported 6 females in the age of…” should perhaps replace ‘females’ with ‘women’.
8. Results, page 3, line 97: I think a full stop is missing at the end of one sentence and the beginning of the next (“(Table I). However…”).
9. Results, page 3, lines 97-98: I am not entirely clear what the phrase “advanced endometriosis was more aggressive in the perimenopausal group…” means. I assume it refers to the finding that endometriosis in perimenopausal women tended to be at a higher stage than in postmenopausal women – and higher stage disease might be viewed as ‘advanced’ or ‘aggressive’. Could this perhaps be rephrased, or else explained (and similarly for the sentence in the abstract)?
10. Results, page 3, line 99: the authors give the percentage of 98/152 as “65, 2%”, but I believe it should be 64.5% (as also given in Table 1).
11. Results, page 3, line 111: “…24 women in postmenopausal group…” should perhaps be “…24 women in the postmenopausal group…”
12. Table I and Table II: The authors don’t necessarily have to repeat the denominator in each cell (i.e. 184 in Group 1, and 46 in Group 2), as this is listed in the top line of the table, and is the same throughout.
13. Table I: Age (years) is given, I think, as a mean, standard deviation, and possibly a range – but this is not specified. There is a note at the end of Table II, which is perhaps also meant to apply to Table I? In any case, it might be helpful to specify in the line of the table.
14. Table II: In the first row (Endometrioma), “125/184 (68%)” is in italics for no clear reason. [I am also uncertain why the percentage is rounded, when others in the table are given to one decimal place].
15. Discussion, page 4, lines 115-116: The first sentence of the discussion has slightly odd phrasing, and might benefit from being re-written.
16. Discussion, page 5, line 159: “Upon now” is an unusual expression; I would suggest perhaps “To date”, “Previously”, or even just starting the sentence at “Genetic mutations…”
17. Discussion, page 5, lines 167-170: The authors correctly describe mutations which have been associated with the endometrioid and clear cell subtypes of ovarian cancer, which may arise from endometriosis in some cases. However, I am uncertain of the justification for the stated link to perimenopausal endometriosis in particular (especially as neither of the papers cited seems to focus particularly on perimenopausal endometriosis).
Author Response
Response to Reviewer 2 Comments
1. Comments and Suggestions for Authors
This manuscript describes the patient demographics and disease features of 184 perimenopausal and 46 postmenopausal women with endometriosis, drawn from a population of 1100 women diagnosed with endometriosis in New Haven (USA) and Greece. The authors compared features between the peri- and post-menopausal women, and found some differences in disease presentation (e.g. perimenopausal women generally had higher stage endometriosis, and were more likely to have endometriomas and uterine leiomyomata, than postmenopausal women).
This paper is useful in providing data on endometriosis in a relatively under-studied population, as endometriosis is often viewed as a disease more of young, pre-menopausal women. The authors also discuss the important clinical concern that endometriosis carries a risk of malignant transformation, which other studies have suggested may be more common in older, peri- or post-menopausal women, and describe endometriosis-associated malignancies in their population.
Response: We are very much thankful to the reviewer for the deep and thorough review. We hope that our revision has improved the paper to a level of his/her satisfaction.
2. General comments
Some general points which I think the authors may which to address are:
Rounding of numbers: Percentages in tables and in the text are mostly given to one decimal place, but on occasion are rounded to the nearest whole number. Ideally, one should be consistent (e.g. either round to the nearest whole number, or give to one decimal place, but do the same throughout).
Decimal points: The authors frequently use a comma (,) rather than a point (.) to separate decimal places (e.g. 18, 4% rather than 18.4%, or p<0,01 rather than p<0.01). While use of a comma in this situation is common in Europe, I believe it is less common in the USA, and so it might be clearer for an international audience to use a point.
Response: We tried to improve and modify the content of the text according to the reviewers’ recommendation.
3. Discussion: In addition to putting the findings of this study into the context of other papers (which the authors do here), it might also be helpful to discuss some potential limitations of the study. For example, to consider possible confounding or biases in the population of patients used here: these are cases identified from clinical chart review, and with a histological diagnosis of endometriosis. These women thus will not be entirely representative of all women with endometriosis in the community (e.g. they may have had some other reason to bring them to hospital and have surgery, such as the cancers or other gynaecological conditions described here). Hence, this study cannot, for example, provide data on the incidence of endometriosis-associated cancers amongst women with endometriosis in the general population.
Response: In our case, we found 184 patients (16. 7%) in the perimenopause group and 46 (4. 2%) in the postmenopausal group. Even though these data are consistent with the observations of previous studies and indicate that physicians should be aware of the possibility of endometriosis and endometriosis associated ovarian cancer in postmenopausal age, the fact that we cannot know if cancer and endometriosis coincided poses a limitation.
4. Conclusion: It is not clear to me that the results described in this paper provide evidence for all the conclusions stated here. In some cases, I think this may simply be suboptimal phrasing. For example: “Our results support an association between endometriosis with perimenopausal and postmenopausal age”; I think the authors may simply mean: “Our results provide further evidence that endometriosis may occur in women of perimenopausal and postmenopausal age.” The authors then state: “Postmenopausal endometriosis infers a risk of malignant transformation such as endometriosis associated ovarian cancer especially in postmenopausal patients.” While it is true that endometriosis carries a risk of malignant transformation (as has been shown by previous papers), and some previous papers have suggested that the risk of malignant transformation may be higher in postmenopausal women (e.g. the paper by DePriest et al 1992), the potential confounding effect of age should perhaps be considered. Postmenopausal women are almost by definition older, and ovarian cancer is more common with increasing age. I’m not clear if there is any evidence that postmenopausal women with endometriosis have a higher rate of malignant transformation than younger women with endometriosis, after adjusting for age? If the authors are aware of such a study, it would be good to cite it (this reviewer may simply have missed it). Finally, the authors state “A larger study would be required to confirm the correlation of endometriosis and menopause”: I am not entirely certain what ‘correlation’ this refers to. If it is simple co-existence, then I think this has been shown (by this paper and others). The term ‘correlation’ perhaps more usually implies a statistical association, in which two factors increase or decrease together (e.g. a correlation between smoking and increasing rates of lung cancer, or between use of the oral contraceptive pill and decreasing rates of ovarian cancer), and I don’t think this paper shows or cites a correlation as such between endometriosis and the menopause. I think the authors may perhaps mean more generally ‘the relationship between endometriosis and menopause’?
Response: We modified the entire conclusion part.
5. References: the authors appear to have correctly numbered the references at the end of the manuscript in citation order, but have used ‘(Author, Date)’ to cite them within the text. However, I believe the reference style for this journal is to number the references in square brackets. Some references are also given in non-standard style, with variable underlining etc. This would need to be corrected prior to publication.
Response: As suggested, we modified and rephrased the references according to the journals’ guidelines.
More minor specific comments
Various more minor specific comments are below. These are mostly to do with phrasing, and the authors may obviously modify as they think best.
1. Introduction, page 1, line 37: “reproductive female population” might be better phrased as “women of reproductive age”.
Response: As suggested, we rephrased the sentence.
2. Introduction, page 2, line 45: “or because the presence of endogenous hormones” should perhaps be either “or because of the presence of endogenous hormones” or “or due to the presence of endogenous hormones”.
Response: As suggested, we rephrased the sentence.
3. Introduction, page 2, line 51: “In postmenopause the real incidence of endometriosis is limited and not consistent” should perhaps be rephrased, e.g. “In postmenopausal women, evidence for the real incidence of endometriosis is limited and not consistent”.
Response: As suggested, we rephrased the sentence.
4. Materials and Methods, page 2, lines 77-79: It is not entirely clear to me why some of this sentence is italicised. Perhaps this was not intentional?
Response: As suggested, we corrected the mistake.
5. Materials and Methods, page 2, lines 82-85: some of the phrasing of the ethics statement seems a little odd. Assuming this still accurately reflects the approvals given, might it be clearer to say: “This study was approved by the Human Committee of Yale University School of Medicine (HIIC #12590), the Ethics Committee for Human Research of Venizeleio Hospital (ECHR#46/6686 and 47/773/2017), the Department of Obstetrics and Gynecology of the University of Crete, and the Department of Obstetrics and Gynecology of the Venizeleio General Hospital of Crete.” [Though obviously please modify if necessary].
Response: As suggested, we rephrased the sentence.
6. Results, page 3, line 95: I think a full stop (.) is missing at the end of one sentence and the beginning of the next (“(Table 1). In the…”).
Response: As suggested, we corrected the mistake.
7. Results, page 3, line 96: “we reported 6 females in the age of…” should perhaps replace ‘females’ with ‘women’.
Response: As suggested, the word is changed.
8. Results, page 3, line 97: I think a full stop is missing at the end of one sentence and the beginning of the next (“(Table I). However…”).
Response: As suggested, we corrected the mistake.
9. Results, page 3, lines 97-98: I am not entirely clear what the phrase “advanced endometriosis was more aggressive in the perimenopausal group…” means. I assume it refers to the finding that endometriosis in perimenopausal women tended to be at a higher stage than in postmenopausal women – and higher stage disease might be viewed as ‘advanced’ or ‘aggressive’. Could this perhaps be rephrased, or else explained (and similarly for the sentence in the abstract)?
Response: Advanced stage means stage III, IV; it is included in the text in the revised version, as suggested.
10. Results, page 3, line 99: the authors give the percentage of 98/152 as “65, 2%”, but I believe it should be 64.5% (as also given in Table 1).
Response: As suggested, we corrected the mistake.
11. Results, page 3, line 111: “…24 women in postmenopausal group…” should perhaps be “…24 women in the postmenopausal group…”
Response: As suggested, we corrected the mistake.
12. Table I and Table II: The authors don’t necessarily have to repeat the denominator in each cell (i.e. 184 in Group 1, and 46 in Group 2), as this is listed in the top line of the table, and is the same throughout.
Response: As suggested, we modified the content.
13. Table I: Age (years) is given, I think, as a mean, standard deviation, and possibly a range – but this is not specified. There is a note at the end of Table II, which is perhaps also meant to apply to Table I? In any case, it might be helpful to specify in the line of the table.
Response: As suggested, we corrected the mistake.
14. Table II: In the first row (Endometrioma), “125/184 (68%)” is in italics for no clear reason. [I am also uncertain why the percentage is rounded, when others in the table are given to one decimal place].
Response: As suggested, we corrected the mistake.
15. Discussion, page 4, lines 115-116: The first sentence of the discussion has slightly odd phrasing, and might benefit from being re-written.
Response: As suggested, we modified the sentence.
16. Discussion, page 5, line 159: “Upon now” is an unusual expression; I would suggest perhaps “To date”, “Previously”, or even just starting the sentence at “Genetic mutations…”
Response: As suggested, we corrected the word.
17. Discussion, page 5, lines 167-170: The authors correctly describe mutations which have been associated with the endometrioid and clear cell subtypes of ovarian cancer, which may arise from endometriosis in some cases. However, I am uncertain of the justification for the stated link to perimenopausal endometriosis in particular (especially as neither of the papers cited seems to focus particularly on perimenopausal endometriosis).
Response: The co existence of endometriosis with various cancers should be linked to the genetic component in the future, although the current literature is limited.
We strongly believe that these ‘’More minor specific comments’’ are very important.

Round 2
Reviewer 1 Report
The manuscript has been significantly modified.
Abstract
- In conclusions part of Abstract and Conclusions section it is mentioned hormonal therapy but neither in materials and methods nor in Results sections is reported that the population in study is treated with hormonal therapy; additionally it is not specified the type (i.e. GnRHa, progestogens or combined estrogen-progestogens) or duration of therapy.
The Authors didn’t mention hormonal therapy anymore neither in the Abstract nor in the Conclusion sections.
-The aim of the study is not well expressed, and incomplete in comparison to the aim described in the manuscript.
Changes have been made regarding the aim of the study
-It is reported that the risk of EAEOC is just in postmenopausal women whereas in “Results” section and in Table 2 it is stated that it has the same incidence also in perimenopausal women.
They stated that the risk of EAEOC is the same in perimenopausal and postmenopausal women.
Introduction
-I suggest to investigate other factors influencing endometriosis associated pain such as 25 OH Vitamin D status, as reported in this recent article “ Anastasi E, Fuggetta E, De Vito C, Migliara G, Viggiani V, Manganaro L, Granato T, Benedetti Panici P, Angeloni A, Porpora MG. Low levels of 25-OH vitamin D in women with endometriosis and associated pelvic pain. Clin Chem Lab Med. 2017 Oct 26;55(12):e282-e284”
They added the recent article” Anastasi E, Fuggetta E, De Vito C, Migliara G, Viggiani V, Manganaro L, Granato T, Benedetti Panici P, Angeloni A, Porpora MG. Low levels of 25-OH vitamin D in women with endometriosis and associated pelvic pain. Clin Chem Lab Med. 2017 Oct 26;55(12):e282-e284”
-The reference Gonzales (2009) should be better discussed.
The reference Gonzales (2009) has been removed.
Material and methods
-The size of the two is too heterogeneous
-Precise inclusion and exclusion criteria should be provided.
Even if the size of the two groups is still heterogeneous, they added inclusion and exclusion criteria and added the timing of biopsy.
Results
-Non statistically significant results shouldn’t be discussed.
-It is not clear why bowel cancer and other types of cancer are included in the table described as “gynecologic conditions”.
Even if they decided to discuss also non statistically significant results, they modified Table II title in order to include also bowel cancer.
Discussion
The Discussion section should start with key result of the study and after that authors should comment and discuss their results in comparison with results of similar studies. I suggest to the authors to rewrite this whole section because there are parts that are not directly associated with this study or should be included in Introduction not Discussion.
The Authors rewrote Discussion section that in the last version starts with key result of the study.
-It is not clear why among the different clinical manifestations of menopause you chose to discuss about DED (dry eye disease), since in your study you didn’t find any correlation with endometrosis and DED is mentioned only as one of the main indication for hormonal replacement therapy. (lines 144-151)
In line 153 they clarify that menopausal status and hormonal disturbance menopause-related can affect different organs, including eyes.
-Additionally, the genetic mutations reported link perimenopause and postmenopause to other clinical conditions (cardiovascular disease or spinal bone loss) than endometriosis, which is the object of the study. (lines 159-163).
They didn’t include the statements about genetic mutations.
-Lines 126; 163-164, the claims should be corroborated by references.
They added references to corroborate their claims at lines 168-170.
-It could be interesting to discuss the underlying mechanism for the presence of adenomyosis in postmenopausal endometriosis and to explain why endometrioma is mostly present in perimenopausal endometriosis.
The underlying mechanism for the presence of adenomyosis in postmenopausal endometriosis has been explained relating endometriosis to extra ovarian estrogen production.
-There is a lack of figures and the tables are poorly designed and don’t report all the discussed data (i.e. in table 2 dried eye symptoms are not reported, even if discussed)
They added dried eye symptoms in Table II.
Conclusion
-All conclusions should be based solely on the obtained result which, in this study, is not the case.
-The sentence “ postmenopausal endometriosis infers a risk of malignant transformation such as endometriosis associated ovarian cancer especially in postmenopausal patients.” doesn’t make sense.
They didn’t stated that “postmenopausal endometriosis infers a risk of malignant transformation such as endometriosis associated ovarian cancer especially in postmenopausal patients” anymore.
-It has not been demonstrated that postmenopausal endometriosis infers a risk of malignant transformation because in the Results section it is stated that the incidence of EAOC is similar into groups.
They stated that the risk of malignant transformation is the same in the 2 studied groups.
-You should specify which type of association you found between endometriosis and perimenopausal and postmenopausal age.
-Moreover, the effects of hormonal replacement therapy in patients with previous endometriosis are mentioned again, but these results were not demonstrated in this study.
- They didn’t mention the effects of hormonal replacement therapy in patients with previous endometriosis.
They didn’t mention hormonal replacement therapy at all.
In its present form the paper is suitable to be published in the Journal.

Author Response
The manuscript has been significantly modified.
Response: We greatly appreciate your thoughtful comments that helped improve our work. We trust that all of your comments have been addressed accordingly in our revised manuscript. We further modified some points.
Abstract
- In conclusions part of Abstract and Conclusions section it is mentioned hormonal therapy but neither in materials and methods nor in Results sections is reported that the population in study is treated with hormonal therapy; additionally it is not specified the type (i.e. GnRHa, progestogens or combined estrogen-progestogens) or duration of therapy.
The Authors didn’t mention hormonal therapy anymore neither in the Abstract nor in the Conclusion sections.
Response: As suggested we modified the sentence.
-The aim of the study is not well expressed, and incomplete in comparison to the aim described in the manuscript.
Changes have been made regarding the aim of the study
Response: As suggested we modified the sentence.
-It is reported that the risk of EAEOC is just in postmenopausal women whereas in “Results” section and in Table 2 it is stated that it has the same incidence also in perimenopausal women.
They stated that the risk of EAEOC is the same in perimenopausal and postmenopausal women.
Response: As suggested we modified the sentence.
Introduction
-I suggest to investigate other factors influencing endometriosis associated pain such as 25 OH Vitamin D status, as reported in this recent article “ Anastasi E, Fuggetta E, De Vito C, Migliara G, Viggiani V, Manganaro L, Granato T, Benedetti Panici P, Angeloni A, Porpora MG. Low levels of 25-OH vitamin D in women with endometriosis and associated pelvic pain. Clin Chem Lab Med. 2017 Oct 26;55(12):e282-e284”
They added the recent article” Anastasi E, Fuggetta E, De Vito C, Migliara G, Viggiani V, Manganaro L, Granato T, Benedetti Panici P, Angeloni A, Porpora MG. Low levels of 25-OH vitamin D in women with endometriosis and associated pelvic pain. Clin Chem Lab Med. 2017 Oct 26;55(12):e282-e284”
Response: As suggested we modified the sentence.
-The reference Gonzales (2009) should be better discussed.
The reference Gonzales (2009) has been removed.
Response: we deleted the reference.
Material and methods
-The size of the two is too heterogeneous
-Precise inclusion and exclusion criteria should be provided.
Even if the size of the two groups is still heterogeneous, they added inclusion and exclusion criteria and added the timing of biopsy.
Response: we deleted the sentence regarding ages,because we explain that in the result section.
Results
-Non statistically significant results shouldn’t be discussed.
Response: We aimed to review the epidemiological aspect of the disease pattern of a series of perimenopausal and postmenopausal women with endometriosis. Non statistically significant results are mentioned only in the tables.
-It is not clear why bowel cancer and other types of cancer are included in the table described as “gynecologic conditions”.
Response: The title of the table is modified. Table II. The coexistence of endometriosis with various benign and malignant conditions
Even if they decided to discuss also non statistically significant results, they modified Table II title in order to include also bowel cancer.
Response: we tried to change the sturcture of table II.
Discussion
The Discussion section should start with key result of the study and after that authors should comment and discuss their results in comparison with results of similar studies. I suggest to the authors to rewrite this whole section because there are parts that are not directly associated with this study or should be included in Introduction not Discussion.
The Authors rewrote Discussion section that in the last version starts with key result of the study.
Response: As suggested we modified the discussion section.
-It is not clear why among the different clinical manifestations of menopause you chose to discuss about DED (dry eye disease), since in your study you didn’t find any correlation with endometrosis and DED is mentioned only as one of the main indication for hormonal replacement therapy. (lines 144-151)
In line 153 they clarify that menopausal status and hormonal disturbance menopause-related can affect different organs, including eyes.
Response: We modified the sentence.
-Additionally, the genetic mutations reported link perimenopause and postmenopause to other clinical conditions (cardiovascular disease or spinal bone loss) than endometriosis, which is the object of the study. (lines 159-163).
They didn’t include the statements about genetic mutations.
Response: We modified the sentence.
-Lines 126; 163-164, the claims should be corroborated by references.
They added references to corroborate their claims at lines 168-170.
Response: We modified the sentence.
-It could be interesting to discuss the underlying mechanism for the presence of adenomyosis in postmenopausal endometriosis and to explain why endometrioma is mostly present in perimenopausal endometriosis.
The underlying mechanism for the presence of adenomyosis in postmenopausal endometriosis has been explained relating endometriosis to extra ovarian estrogen production.
Response: We modified the sentence.
-There is a lack of figures and the tables are poorly designed and don’t report all the discussed data (i.e. in table 2 dried eye symptoms are not reported, even if discussed)
They added dried eye symptoms in Table II.
Response : we tried to change the sturcture of table II, moreover we added p value for Dry eye.
Conclusion
-All conclusions should be based solely on the obtained result which, in this study, is not the case.
Response: We modified this section.
-The sentence “ postmenopausal endometriosis infers a risk of malignant transformation such as endometriosis associated ovarian cancer especially in postmenopausal patients.” doesn’t make sense.
They didn’t stated that “postmenopausal endometriosis infers a risk of malignant transformation such as endometriosis associated ovarian cancer especially in postmenopausal patients” anymore.
Response: We modified this section as suggested.
-It has not been demonstrated that postmenopausal endometriosis infers a risk of malignant transformation because in the Results section it is stated that the incidence of EAOC is similar into groups.
They stated that the risk of malignant transformation is the same in the 2 studied groups.
Response: We modified this section.
-You should specify which type of association you found between endometriosis and perimenopausal and postmenopausal age.
Response: Our results support an association between endometriosis with perimenopausal and postmenopausal age. We detected that endometrioma, uterine leiomyomata and advanced endometriosis are more prominent in perimenopausal group, compared to the postmenopausal group, where adenomyosis was found in a higher rate. Apart from that, we confirmed an association between endometriosis and endometriosis- associated ovarian cancer in both groups.
-Moreover, the effects of hormonal replacement therapy in patients with previous endometriosis are mentioned again, but these results were not demonstrated in this study.
- They didn’t mention the effects of hormonal replacement therapy in patients with previous endometriosis.
They didn’t mention hormonal replacement therapy at all.
Response: We did not include hormonal replacement therapy at all.
In its present form the paper is suitable to be published in the Journal.